# Genotype Value Decomposition: Simple Methods for the Computation of Kernel Statistics

*Kazuharu Misawa*

Recent advances in sequencing technologies enable genome-wide analyses for thousands of individuals. The sequential kernel association test (SKAT) is a widely used method to test for associations between a phenotype and a set of rare variants. As the sample size of human genetics studies increases, the computational time required to calculate a kernel is becoming more and more problematic. In this study, a new method to obtain kernel statistics without calculating a kernel matrix is proposed. A simple method for the computation of two kernel statistics, namely, a kernel statistic based on a genetic relationship matrix (GRM) and one based on an identity by state (IBS) matrix, are proposed. By using this method, calculation of the kernel statistics can be conducted using vector calculation without matrix calculation. The proposed method enables one to conduct SKAT for large samples of human genetics.

## 1. Introduction

A very large number of human genome sequences are now available for the study of human genetics, because of recent advances in genome sequencers. A recent study has shown that rare variants substantially contribute to phenotype variation.[1] Because each linkage disequilibrium block can be analyzed independently, increases in the number of sites can be tackled with parallel computation.[2] However, the statistical power of classical single-marker association analysis for rare variants is quite limited.

To address this challenge, rare and low-frequency variants are often grouped into gene or pathway levels, and the effects of multiple variants evaluated are based on collapsing methods.[3,4] The sequential kernel association test (SKAT)[5,6] is one such popular method. SKAT applies a test statistic $S$, which is defined by the quadratic form, $S = \gamma^T K \gamma$, where $\gamma$ is column vector of the phenotype defined by Equation (1).

$$\mathbf{y} = (\gamma(1) \quad \gamma(2) \quad \cdots \quad \gamma(n))^T \quad (1)$$

K. Misawa
Department of Human Genetics
Yokohama City University Graduate School of Medicine
3-9 Fukuura, Kanazawa-ku, Yokohama 236-0004, Japan
E-mail: kazu_miawa@hotmail.com

 The ORCID identification number(s) for the author(s) of this article can be found under https://doi.org/10.1002/ggn2.202100066

where $\gamma(i)$ is the phenotype value of the $i$-th individual and $n$ is the sample size. In the following section, we assume the average of the elements of $\gamma$ is 0.

Evaluating the probability density of the null distribution of $S$ is important for conducting SKAT, but it requires computing a matrix related to the genotype covariance between markers, which requires a very long computational time. When the length of $\gamma$ is $n$ and the size of the matrix $\mathbf{K}$ is $n^2$. For example, when the number of people is 10 000, the size of the kernel matrix is 100 000 000. A genetic relationship matrix (GRM) among individuals is used in genome-wide complex trait analysis[7] and in principal component analysis.[8] Identity by state (IBS) defines similarity between individuals as the number of shared alleles. The IBS kernel is used in linear regression[9,10] and SKAT.[5]

The aim of the present study is to develop simple methods for the computation of these two kernel statistics without calculating a GRM and an IBS matrix explicitly.

## 2. Theory

### 2.1. Genotype Value Vectors

In the present study, all sites are assumed to be biallelic, namely, each site has a reference allele and an alternative allele. Let us define genotype value vectors at site $k$. Let $a_k(i)$ be 1 when the individual $i$ is a homozygote of a reference allele at site $k$, otherwise $a_k(i) = 0$. Let $b_k(i)$ be 1 when the individual $i$ is the heterozygote at site $k$, otherwise $b_k(i) = 0$. Let $c_k(i)$ be 1 when the individual $i$ is a homozygote of an alternative allele at site $k$, otherwise $c_k(i) = 0$. The vectors $a_k, b_k, c_k$ are defined by Equation (2).

$$\mathbf{a}_k = (\ a_k(1)\ a_k(2)\ \cdots\ a_k(n)\ )^T$$

$$\mathbf{b}_k = (\ b_k(1)\ b_k(2)\ \cdots\ b_k(n)\ )^T$$

$$\mathbf{c}_k = (\ c_k(1)\ c_k(2)\ \cdots\ c_k(n)\ )^T \quad (2)$$

Let us denote $a_k$, $b_k$, and $c_k$ as the genotype value vectors. Because $a_k(i) + b_k(i) + c_k(i) = 1$, it is worth noting that

$$\mathbf{a}_k + \mathbf{b}_k + \mathbf{c}_k = \mathbf{1} \quad (3)$$

where $\mathbf{1}$ is defined by $\mathbf{1} = (\ 1\ 1\ \cdots\ 1\ )^T$ The alternative allele frequency, $p$, is obtained by $p = (\mathbf{b}_k^T \mathbf{1} + 2\mathbf{c}_k^T \mathbf{1})/(2n)$. In the follow-

**Table 1.** Relationship between genotype values and the number of alternative alleles.

| Genotype | $a_k(i)$ | $b_k(i)$ | $c_k(i)$ | $g_k(i)$ |
|---|---|---|---|---|
| 0/0 | 1 | 0 | 0 | 0 |
| 0/1 | 0 | 1 | 0 | 1 |
| 1/1 | 0 | 0 | 1 | 2 |

ing section, the Hardy–Weinberg equilibrium is assumed for this site. Namely, the frequencies of heterozygotes and homozygotes of the alternative alleles are $2p(1 − p)$ and $p^2$, respectively.

## 2.2. The GRM Kernel

The allele values are 0 for the reference allele, and 1 for the alternative allele. The separator between the alleles is "/" as used in the variant call format.[11] Let $g_k(i)$ be the genotype value for individual $i$ at site $k$. In the present study, $g_k(i)$ is the number of alternative alleles. The relationship between the genotype of the individual $i$ at site $k$ and $g_k(i)$ is shown in **Table 1**. The vector $\mathbf{g}_k$ is defined by Equation (4).

$$\mathbf{g}_k = \begin{pmatrix} g_k(1) & g_k(2) & \cdots & g_k(n) \end{pmatrix}^T \tag{4}$$

Table 1 displays the relationships among allelic states $a_k(i)$, $b_k(i)$, $c_k(i)$, and $g_k(i)$. Because $g_k(i) = b_k(i) + 2c_k(i)$, $\mathbf{g}_k$ is obtained by $\mathbf{g}_k = \mathbf{b}_k + 2\mathbf{c}_k$.

Let us denote the GRM at site $k$ as $\mathbf{X}_k$. We subtract the mean $\mu_k = \{\sum_{i=1}^n g_k(i)\}/n$ to obtain a matrix with row sums equal to 0. The $ij$-th element of $\mathbf{X}_k$ at site $k$ is obtained using Equation (5).

$$\mathbf{X}_k(i,j) = \{g_k(i) − \mu_k\}\{g_k(j) − \mu_k\}$$
$$= g_k(i)g_k(j) − \mu_k g_k(i) − \mu_k g_k(j) + \mu_k^2 \tag{5}$$

$$\mathbf{X}_k = \begin{pmatrix} g_k(1)g_k(1) & g_k(1)g_k(2) & \cdots & g_k(1)g_k(n) \\ g_k(2)g_k(1) & g_k(2)g_k(2) & \cdots & g_k(2)g_k(n) \\ \vdots & \vdots & & \vdots \\ g_k(n)g_k(1) & g_k(n)g_k(2) & \cdots & g_k(n)g_k(n) \end{pmatrix}$$
$$- \mu_k \begin{pmatrix} g_k(1) & g_k(1) & \cdots & g_k(1) \\ g_k(2) & g_k(2) & \cdots & g_k(2) \\ \vdots & \vdots & & \vdots \\ g_k(n) & g_k(n) & \cdots & g_k(n) \end{pmatrix}$$
$$- \mu_k \begin{pmatrix} g_k(1) & g_k(2) & \cdots & g_k(n) \\ g_k(1) & g_k(2) & \cdots & g_k(n) \\ \vdots & \vdots & & \vdots \\ g_k(1) & g_k(2) & \cdots & g_k(n) \end{pmatrix} + \mu_k^2 \begin{pmatrix} 1 & 1 & \cdots & 1 \\ 1 & 1 & \cdots & 1 \\ \vdots & \vdots & & \vdots \\ 1 & 1 & \cdots & 1 \end{pmatrix} \tag{6}$$

Subsequently, the matrix $\mathbf{X}_k$ can be obtained using the genotype value vectors.

Let us define a new matrix $\mathbf{G}_k$. As shown in Table 1, the $ij$-th element of $\mathbf{G}_k$ at site $k$ can be calculated using Equation (7).

$$\mathbf{G}_k(i,j) = g_k(i)g_k(j) \tag{7}$$

Subsequently, the matrix $\mathbf{G}_k$ would be obtained using the genotype value vectors.

$$\mathbf{G}_k = \begin{pmatrix} g_k(1)g_k(1) & g_k(1)g_k(2) & \cdots & g_k(1)g_k(n) \\ g_k(2)g_k(1) & g_k(2)g_k(2) & \cdots & g_k(2)g_k(n) \\ \vdots & \vdots & & \vdots \\ g_k(n)g_k(1) & g_k(n)g_k(2) & \cdots & g_k(n)g_k(n) \end{pmatrix}$$
$$= \begin{pmatrix} g_k(1) \\ g_k(2) \\ \vdots \\ g_k(n) \end{pmatrix} \begin{pmatrix} g_k(1) & g_k(2) & \cdots & g_k(n) \end{pmatrix} = \mathbf{g}_k \mathbf{g}_k^T$$
$$\begin{pmatrix} g_k(1) & g_k(1) & \cdots & g_k(1) \\ g_k(2) & g_k(2) & \cdots & g_k(2) \\ \vdots & \vdots & & \vdots \\ g_k(n) & g_k(n) & \cdots & g_k(n) \end{pmatrix} = \mathbf{1}\mathbf{g}_k^T$$
$$\begin{pmatrix} g_k(1) & g_k(2) & \cdots & g_k(n) \\ g_k(1) & g_k(2) & \cdots & g_k(n) \\ \vdots & \vdots & & \vdots \\ g_k(1) & g_k(2) & \cdots & g_k(n) \end{pmatrix} = \mathbf{g}_k \mathbf{1}^T$$
$$\begin{pmatrix} 1 & 1 & \cdots & 1 \\ 1 & 1 & \cdots & 1 \\ \vdots & \vdots & & \vdots \\ 1 & 1 & \cdots & 1 \end{pmatrix} = \mathbf{1}\mathbf{1}^T \tag{8}$$

Therefore, we obtain Equation (9).

$$\mathbf{X}_k = \mathbf{G}_k − \mu \mathbf{1}\mathbf{g}_k^T − \mu \mathbf{g}_k \mathbf{1}^T + \mu^2 \mathbf{1}\mathbf{1}^T \tag{9}$$

It is worth noting that

$$\mathbf{y}^T \mathbf{X}_k \mathbf{y} = \mathbf{y}^T \mathbf{G}_k \mathbf{y} + \mu \mathbf{y}^T \mathbf{1}\mathbf{g}_k^T \mathbf{y} + \mu \mathbf{y}^T \mathbf{g}_k \mathbf{1}^T \mathbf{y} + \mu^2 \mathbf{y}^T \mathbf{1}\mathbf{1}^T \mathbf{y} \tag{10}$$

Because $\mathbf{y}^T \mathbf{1} = \mathbf{1}^T \mathbf{y} = 0$, we obtain

$$\mathbf{y}^T \mathbf{X}_k \mathbf{y} = \mathbf{y}^T \mathbf{G}_k \mathbf{y} \tag{11}$$

By using the distributivity and associativity of matrix production, we obtain

$$\mathbf{G}_k = (\mathbf{b}_k + 2\mathbf{c})(\mathbf{b}_k + 2\mathbf{c}_k)^T \tag{12}$$

$Q_k$ is a scalar value of site $k$ defined by Equation (13):

$$Q_k = \mathbf{y}^T \mathbf{G}_k \mathbf{y} = \mathbf{y}^T (\mathbf{b}_k + 2\mathbf{c}_k)(\mathbf{b}_k + 2\mathbf{c}_k)^T \mathbf{y} = (\mathbf{y}^T \mathbf{b}_k + 2\mathbf{y}^T \mathbf{c}_k)^2 \tag{13}$$

because the transpose of a product of matrices is the product, in the reverse order, of the transposes of the factors. Note $\mathbf{y}^T \mathbf{b}_k + 2\mathbf{y}^T \mathbf{c}_k$ is a scalar that can be obtained as

$$\mathbf{y}^T \mathbf{b}_k + 2\mathbf{y}^T \mathbf{c}_k = \sum_{i=1}^n y(i)\{b_k(i) + 2c_k(i)\} \tag{14}$$

## 2.3. The IBS Kernel

IBS defines similarity between individuals as the number of shared alleles. The IBS kernel is used in linear regression[9,10]

**2100066 (2 of 5)**

**Table 2.** Relationship between genotype values and identities by state (IBS).

| Individual $i$ | | | | Individual $j$ | | | | |
|---|---|---|---|---|---|---|---|---|
| Genotype | $a_k(i)$ | $b_k(i)$ | $c_k(i)$ | Genotype | $a_k(j)$ | $b_k(j)$ | $c_k(j)$ | IBS |
| 0/0 | 1 | 0 | 0 | 0/0 | 1 | 0 | 0 | 2 |
| 0/0 | 1 | 0 | 0 | 0/1 | 0 | 1 | 0 | 1 |
| 0/0 | 1 | 0 | 0 | 1/1 | 0 | 0 | 1 | 0 |
| 0/1 | 0 | 1 | 0 | 0/0 | 1 | 0 | 0 | 1 |
| 0/1 | 0 | 1 | 0 | 0/1 | 0 | 1 | 0 | 2 |
| 0/1 | 0 | 1 | 0 | 1/1 | 0 | 0 | 1 | 1 |
| 1/1 | 0 | 0 | 1 | 0/0 | 1 | 0 | 0 | 0 |
| 1/1 | 0 | 0 | 1 | 0/1 | 0 | 1 | 0 | 1 |
| 1/1 | 0 | 0 | 1 | 1/1 | 0 | 0 | 1 | 2 |

**Table 3.** SNPs used in the computer simulation.

| Chromosome | Position on hg19 | rsID |
|---|---|---|
| 11 | 64360996 | rs552232030 |
| 11 | 64361124 | rs201136391 |
| 11 | 64361219 | rs121907892 |
| 11 | 64366298 | rs150255373 |
| 11 | 64367290 | rs563239942 |
| 11 | 64368212 | rs200104135 |
| 11 | 64368968 | rs528619562 |

and the SKAT.[5] Let $\mathrm{IBS}_k(i)$ be the $ij$-th element of the IBS matrix, $\mathbf{IBS}_k$, at site $k$, which denotes the number of shared alleles by subjects $i$ and $j$ at site $k$.

**Table 2** displays the relationships between genotype values and IBS. From the table, we can observe the following relationship among genotype value vectors and the IBS matrix.

$$\mathrm{IBS}_k\ (i,j) = 2a_k(i)\,a_k(j) + b_k(i) + b_k(j) + 2c_k(i)\,c_k(j) \tag{15}$$

Thus, the IBS matrix at site $k$ is obtained by

$$\mathbf{IBS}_k = 2\begin{pmatrix} a_k(1)\,a_k(1) & a_k(1)\,a_k(2) & \cdots & a_k(1)\,a_k(n) \\ a_k(2)\,a_k(1) & a_k(2)\,a_k(2) & \cdots & a_k(2)\,a_k(n) \\ \vdots & \vdots & & \vdots \\ a_k(n)\,a_k(1) & a_k(n)\,a_k(2) & \cdots & a_k(n)\,a_k(n) \end{pmatrix}$$

$$+ \begin{pmatrix} b_k(1) & b_k(1) & \cdots & b_k(1) \\ b_k(2) & b_k(2) & \cdots & b_k(2) \\ \vdots & \vdots & & \vdots \\ b_k(n) & b_k(n) & \cdots & b_k(n) \end{pmatrix}$$

$$+ \begin{pmatrix} b_k(1) & b_k(2) & \cdots & b_k(n) \\ b_k(1) & b_k(2) & \cdots & b_k(n) \\ \vdots & \vdots & & \vdots \\ b_k(1) & b_k(2) & \cdots & b_k(n) \end{pmatrix}$$

$$+ 2\begin{pmatrix} c_k(1)\,c_k(1) & c_k(1)\,c_k(2) & \cdots & c_k(1)\,c_k(n) \\ c_k(2)\,c_k(1) & c_k(2)\,c_k(2) & \cdots & c_k(2)\,c_k(n) \\ \vdots & \vdots & & \vdots \\ c_k(n)\,c_k(1) & c_k(n)\,c_k(2) & \cdots & c_k(n)\,c_k(n) \end{pmatrix}$$

$$= 2\mathbf{a}_k\mathbf{a}_k^{\mathrm{T}} + 1\mathbf{b}_k^{\mathrm{T}} + \mathbf{b}_k 1^{\mathrm{T}} + 2\mathbf{c}_k\mathbf{c}_k^{\mathrm{T}} \tag{16}$$

$R_k$ is a scalar value of site $k$ defined by Equation (17).

$$R_k = \mathbf{y}^{\mathrm{T}}\mathbf{IBS}_k\mathbf{y} \tag{17}$$

By using the distributivity and associativity of matrix production, we obtain

$$R_k = \mathbf{y}^{\mathrm{T}}\left(\mathbf{a}_k\mathbf{a}_k^{\mathrm{T}} + 1\mathbf{b}_k^{\mathrm{T}} + \mathbf{b}_k 1^{\mathrm{T}} + \mathbf{c}_k\mathbf{c}_k^{\mathrm{T}}\right)\mathbf{y}$$

$$= 2\mathbf{y}^{\mathrm{T}}\left(\mathbf{a}_k\mathbf{a}_k^{\mathrm{T}}\right)\mathbf{y} + \mathbf{y}^{\mathrm{T}}1\mathbf{b}_k^{\mathrm{T}}\mathbf{y} + \mathbf{y}^{\mathrm{T}}\mathbf{b}_k 1^{\mathrm{T}}\mathbf{y} + 2\mathbf{y}^{\mathrm{T}}\left(\mathbf{c}_k\mathbf{c}_k^{\mathrm{T}}\right)\mathbf{y}$$

$$= 2\left(\mathbf{y}^{\mathrm{T}}\mathbf{a}_k\right)^2 + 2\left(\mathbf{y}^{\mathrm{T}}\mathbf{c}_k\right)^2 \tag{18}$$

$\mathbf{y}^{\mathrm{T}}\mathbf{a}_k$ and $\mathbf{y}^{\mathrm{T}}\mathbf{c}_k$ are scalars that can be obtained using the inner product of two vectors. By using Equation (3), we can obtain $\mathbf{y}^{\mathrm{T}}\mathbf{a}_k = \mathbf{y}^{\mathrm{T}}1 - \mathbf{y}^{\mathrm{T}}\mathbf{b}_k - \mathbf{y}^{\mathrm{T}}\mathbf{c}_k$.

When multiple single-nucleotide polymorphisms (SNPs) are investigated, the entire GRM and IBS matrices are obtained using $S = \sum_{k=1}^{l} w_k Q_k$ and $S = \sum_{k=1}^{l} w_k R_k$, respectively, where $w_k$ is weight of site $k$ and $l$ is the number of sites. SKAT allows the incorporation of flexible weight functions.[12] Weights can normalize each data column to have the same variance[13] and can increase the power of tests.[13]

### 2.4. Computer Simulations

To evaluate the new method, I performed computer simulations. The python scripts used in the computer simulation are shown in Material 1, Supporting Information. The usages are in Material 2, Supporting Information. This program is ready for data analysis.

#### 2.4.1. Genotype Selection

SNPs on the SLC22A2 gene that are known to affect uric acid levels[1,14–16] were selected. Then, genetic variation of these SNPs of 2504 individuals were downloaded from the 1000 Genomes Project. Monomorphic sites were excluded. As a result, the sites in **Table 3** were used in the computer simulation.

#### 2.4.2. Phenotype Generations

The heterozygous individuals and the homozygous individuals of alternative allele of the uric acid level were set to be 1.0 µg dL⁻¹ lower than the homozygotes of the reference allele. A random variable that follows the normal distribution with mean 0.0 µg dL⁻¹ and standard deviation 1.0 µg dL⁻¹ was added to the uric acid level of each individual in the simulation as an environmental factor of uric acids level. These values are similar to the observed values.[1]

#### 2.4.3. Calculation of Test Statistics and Permutation Tests

For each of these phenotypes, the test statistics of GRM and IBS were observed. Then the permutation tests were performed with 1 000 000 permutations to calculate the probability of exceeding the observed score. The significance level was set to be $5 \times 10^{-6}$, because the number of tests of genome-wide SKAT will be $10^4$. Each permutation test was repeated ten times ($n = 10$).

**2100066 (3 of 5)**

**Table 4.** Power and Computational time of the GRM and IBS tests.

| Method | The number of tests that reject the null hypothesis | Time |
| --- | --- | --- |
| GRM | 2 out of 10 | 1 min 38 s |
| IBS | 3 out of 10 | 1 min 47 s |

## 3. Results

**Table 4** shows that there is no significant difference between the GRM and IBS in the statistical power (the chi-square test, $n = 10$, $P > 5\%$). Table 4 also shows that permutation tests can be conducted in a short period of time by using the methods proposed in this study.

## 4. Discussion

We demonstrate that necessary variant/phenotype association test statistics can be obtained without obtaining eigenvalues and eigenvectors of GRM and IBS matrices, in the present study. The method is referred to as genotype value decomposition. The new methods proposed in this study are conducted with computational time of $O(n)$, where $n$ is the sample size. Notably, these new methods are applicable for common variants as well as rare variants, even though the methods were developed for the association tests for rare variants. Sparse matrix computation can be used when all of variants are rare.

When the alternative allele frequency is very small, homozygotes of the alternative allele are very rare, so that $c_k$ is ignorable. In other words, $Q_k$ can be approximately obtained by $Q_k \approx (\mathbf{y}^\mathrm{T}\mathbf{b}_k)^2$. Under the same condition, $(\mathbf{y}^\mathrm{T}\mathbf{a}_k)^2 \approx (\mathbf{y}^\mathrm{T}\mathbf{b}_k)^2$ and $(\mathbf{y}^\mathrm{T}\mathbf{c}_k)^2 \approx 0$, so that $R_k$ is approximately equal to $2Q_k$.

On one hand, when all sites are independent, the necessary probability density functions can be calculated using convolution of the probability density functions of all sites. On the other hand, it is difficult to obtain convolution of the probability density functions when the sites are linked and dependent on each other. In such cases, a permutation test is used.[6]

Because the statistics calculated by the new method are not approximations but exact values, the null distributions of these statistics are exactly the same as the test statistics with calculating GRM and IBS matrices. Wu et al.[5] showed that the test statistics approximately follow the chi-square distribution. Furthermore, because the distribution is derived from an asymptotic distribution of its statistics, the $p$-values for datasets with an insufficient number of samples may be inaccurate, which could cause inflation or power loss.[] In a permutation test, the test statistic null distribution can be approximated by fully resampling the observed traits without replacement. The proposed method can be useful for reducing computational time to obtain $p$-values using resampling methods.

## 5. Conclusion

In the present paper, a genotype value decomposition method is proposed for handling the kernel matrices. The method can be re-

ferred to as genotype value decomposition. By using this method, calculation of the null distribution of the kernel statistics can be conducted with time complexity $O(n)$. The proposed method enables one to conduct SKAT for large samples of human genetics.

## Supporting Information

Supporting Information is available from the Wiley Online Library or from the author.

## Acknowledgements

The author thanks Dr. Naomichi Matsumoto for his suggestions and encouragement. This work was supported by JSPS KAKENHI Grant Numbers JP17K08682, JP19K22647, JP20K07316. The author also thanks Steven M. Thompson, from Edanz Group for editing a draft of this manuscript.

## Conflict of Interest

The author declares no conflict of interest.

## Data Availability Statement

The python code used in the study is available at https://github.com/kazumisawa/paraHaplo5 under the MIT license.

## Peer Review

The peer review history for this article is available in the Supporting Information for this article.

## Keywords

genetic relationship matrix, identity by state, rare variants, sequential kernel association tests

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

# ADVANCED
## SCIENCE NEWS

www.advancedsciencenews.com

# ADVANCED
# GENETICS

www.advgenet.com

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
