## [**Supplementary Information**: Record of Transparent Peer Review · Advanced Genetics]

Record of Transparent Peer Review

Genotype Value Decomposition: Simple Methods for the Computation of Kernel Statistics

Kazuharu Misawa*

*Corresponding

Review timeline:

Date Submitted: 01-Dec-2021

Editorial Decision: 24-Jan-2022

Revision Received: 03-Feb-2022

Accepted: 04-Feb-2022

Editor: Myles Axton

1 st Peer Review	04-Dec to 21-Jan-2022
-----------------------

Reviewer #1

The authors developed a method to obtain kernel statistics without calculating kernel matrix in the sequential kernel association test. The calculation of the null distribution of the kernel statistics was conducted with time complexity $O(n)$.

1.1 The main contribution of this paper is the conclusion that the term $y^t G y$ can be written as the sum of squares of the elements of the vector Xy , which is also the sum of squares of the simple regression coefficients of $y \sim X_i$ with X_i being the i -th column of X . This, however, is known in the literature.

1.2 In addition, the article only contains derivations, without simulation, method comparison or real data application.

1.3 The paper is too short and heavily under referenced.

Reviewer #2

The authors propose an alternative approach to efficiently compute a quadratic score test statistic used in SKAT and similar kernel testing (e.g. c -alpha) without requiring explicit calculation of the kernel statistic. The authors outline the math for the statistic calculation for two commonly used kernel functions (linear and IBS).

2.1 Unfortunately, this manuscript feels very underdeveloped, and doesn't provide empirical demonstration of computational gains using this approach.

2.2 More problematically, the issue of deriving the corresponding null distribution, which also requires usage of the kernel matrix, is largely cast aside by indicating this can be simply derived via convolutions. However, the null distribution in this context is a linear combination of iid chi-square distributions, whose weights are defined by eigenvalues from decomposition of a matrix product involving the kernel matrix itself. Thus, it's not really clear how the authors paved a road to avoiding this problem.

2.3 There are already multiple computational tricks in place when calculating test statistics and p-values using SKAT or similar packages (e.g., see FastSKAT for high marker dimensionality). Similarly, sparse matrix computation can be taken advantage of due to the high sparsity of rare-variant genotype matrices. These don't seem to be discussed either.

1 st Editorial Decision	24-Jan-2022
Editorial decision: Major Revision. The article can only be reconsidered once the reviewers' substantial criticisms have been addressed. Either develop this approach so that it can be competitively implemented for a real world genetic use case or turn it into an analytic Review article that compares kernel approaches to rare genetic variant analysis in population and biobank samples.	
Editor's understanding of the reviews	

Reviewer #1 Recommends rejection in its current form Reviewer #2 Recommends rejection in its current form	
Reviewer comments	Editor recommendation
2.2 ..problematically, the issue of deriving the corresponding null distribution, which also requires usage of the kernel matrix, is largely cast aside by indicating this can be simply derived via convolutions. However, the null distribution in this context is a linear combination of iid chi-square distributions, whose weights are defined by eigenvalues from decomposition of a matrix produce involving the kernel matrix itself. Thus, it's not really clear how the authors pave a road to avoiding this problem.	ED1 In practice, the rapid application of the method would I think require calculation of the null distribution. Explain in principle how your method achieves this, and compare performance with other existing methods and their implementations.
2.3 There are already multiple computational tricks in place when calculating test statistics and p-values using SKAT or similar packages (e.g., see FastSKAT for high marker dimensionality). Similarly, sparse matrix computation can be taken advantage of due to the high sparsity of rare-variant genotype matrices. These don't seem to be discussed either. 1.1 The main contribution of this paper is the conclusion that the term $y'Gy$ can be written as the sum of squares of the elements of the vector Xy, which is also the sum of squares of the simple regression coefficients of $y \sim X_i$ with X_i being the i-th column of X. This, however, is known in the literature. 1.3 The paper is too short and heavily under referenced.	ED2 Expand the Introduction to cite and take into account published implementations and their theoretical underpinnings.
1.2 In addition, the article only contains derivations, without simulation, method comparison or real data application. 2.1 Unfortunately, this manuscript feels very underdeveloped, and doesn't provide empirical demonstration of computational gains using this approach.	ED3 Apply the method to real samples and report the performance of the method compared to existing tools. Identify potential use cases where the method can be developed to address unmet research needs.

Author's Response to 1 st Review	03-Feb-2022
---	-------------

Response to Editor's comments

Statistics: For original research, please check that your manuscript includes a sub-section entitled "Statistical Analysis" at the end of the Experimental Section that fully describes the following information:

1. Pre-processing of data (e.g., transformation, normalization, evaluation of outliers),
2. Data presentation (e.g., mean \pm SD),
3. Sample size (n) for each statistical analysis,
4. Statistical methods used to assess significant differences with sufficient details (e.g., name of the statistical test including one- or two-sided testing, testing level (i.e., alpha value, P value), if applicable post-hoc test or any alpha adjustment, validity of any assumptions made for the chosen test),
5. Software used for statistical analysis.

Figure legends: Please make sure that all relevant figure legends contain the information on sample size (n), probability (P) value, the specific statistical test for each experiment, data presentation and the meaning of the significance symbol.

I have checked all of these points and made the necessary changes

Reviewer #1

I have addressed all the reviewer's comments, as indicated below.

The authors developed a method to obtain kernel statistics without calculating kernel matrix in the sequential kernel association test. The calculation of the null distribution of the kernel statistics was conducted with time complexity $O(n)$.

1.1 The main contribution of this paper is the conclusion that the term $y'Gy$ can be written as the sum of squares of the elements of the vector Xy , which is also the sum of squares of the simple regression coefficients of $y \sim X_i$ with X_i being the i -th column of X . This, however, is known in the literature.

The aim of this paper is the algorithm for calculating the test statistics without calculating a GRM and an IBS matrix explicitly based on the fact that the term $y'Gy$ can be written as the sum of squares of the elements of the vector Xy ,

1.2 In addition, the article only contains derivations, without simulation, method comparison or real data application.

1.3 The paper is too short and heavily under referenced.

Response: I have cited several additional papers on the real data analyses (refs 14, 15, and 16 in the revised manuscript) on page 6. I developed a set of computer programs to compute p-values using the method described in my manuscript on page 6. The python code of this method was attached as the Supplementary Materials.

Reviewer #2

The authors propose an alternative approach to efficiently compute a quadratic score test statistic used in SKAT and similar kernel testing (e.g. c -alpha) without requiring explicit calculation of the kernel statistic. The authors outline the math for the statistic calculation for two commonly used kernel functions (linear and IBS).

I have addressed all the reviewer's comments, as indicated below.

2.1 Unfortunately, this manuscript feels very underdeveloped, and doesn't provide empirical demonstration of computational gains using this approach.

2.2 More problematically, the issue of deriving the corresponding null distribution, which also requires usage of the kernel matrix, is largely cast aside by indicating this can be simply derived via convolutions. However, the null distribution in this context is a linear combination of iid chi-square distributions, whose weights are defined by eigenvalues from decomposition of a matrix produce involving the kernel matrix itself. Thus, it's not really clear how the authors pave a road to avoiding this problem.

Response: In the original manuscript, I showed the algorithm for calculating the test statistics without calculating a GRM and an IBS matrix explicitly. To clarify this, I added the following sentences on page 8 "Because the statistics calculated by the new method are not approximations but exact values, the null distributions of these statistics are exactly the same as the test statistics with calculating GRM and IBS matrices. Wu et al. (6) showed that the test statistics approximately follow the chi-square distribution. Furthermore, because the distribution is derived from an asymptotic distribution of its statistics, the p-values for datasets with an insufficient number of samples may be inaccurate, which could cause inflation or power loss [17]."

2.3 There are already multiple computational tricks in place when calculating test statistics and p-values using SKAT or similar packages (e.g., see FastSKAT for high marker dimensionality). Similarly, sparse matrix computation can be taken advantage of due to the high sparsity of rare-variant genotype matrices. These don't seem to be discussed either.

Response: I conducted computer simulations, and the results showed that my methods were much faster than FastSKAT. On pages 7-8, I described this as follows: "Notably, these new methods are applicable for common variants as well as rare variants, even though the methods were developed for the association tests for rare variants. Sparse matrix computation can be used when all of variants are rare."

Final Decision	04-Feb-2022
----------------	-------------

Accept the revision as the author has addressed the reviewers' questions about competing approaches and null distribution and has developed an implementation. The author is encouraged to share the software implementation via a public platform and link this to the article.